# Genetic Identification and Transcriptome Analysis of Lintless and Fuzzless Traits in *Gossypium arboreum* L.

**DOI:** 10.3390/ijms21051675

**Published:** 2020-02-29

**Authors:** Xueying Liu, Philippe Moncuquet, Qian-Hao Zhu, Warwick Stiller, Zhengsheng Zhang, Iain Wilson

**Affiliations:** 1CSIRO Agriculture and Food, GPO Box 1700, Canberra ACT 2061, Australia; lxy0322@email.swu.edu.cn (X.L.); Philippe.Moncuquet@csiro.au (P.M.); Qianhao.Zhu@csiro.au (Q.-H.Z.); 2College of Agronomy and Biotechnology, Southwest University, Chongqing 400716, China; zhangzs@swu.edu.cn; 3CSIRO Agriculture and Food, Locked Bag 59, Narrabri NSW 2390, Australia; Warwick.stiller@csiro.au

**Keywords:** fuzzless, lintless, glabrous, fibre development, transcriptome analysis, *Gossypium arboreum*

## Abstract

Cotton fibres, as single cells arising from the seed coat, can be classified as lint and fuzz according to their final length. *Gossypium arboreum* is a cultivated diploid cotton species and a potential donor of the A subgenome of the more widely grown tetraploid cottons. In this study, we performed genetic studies on one lintless and seven fuzzless *G. arboreum* accessions. Through association and genetic linkage analyses, a recessive locus on Chr06 containing *GaHD-1* was found to be the likely gene underlying the lintless trait. *GaHD-1* carried a mutation at a splicing acceptor site that resulted in alternative splicing and a deletion of 247 amino acid from the protein. The regions containing *GaGIR1* and *GaMYB25-like* were found to be associated with fuzz development in *G. arboreum,* with the former being the major contributor. Comparative transcriptome analyses using 0-5 days post-anthesis (dpa) ovules from lintless, fuzzless, and normal fuzzy seed *G. arboreum* accessions revealed gene modules and hub genes potentially important for lint and fuzz initiation and development. Three significant modules and 26 hub genes associated with lint fibre initiation were detected by weighted gene co-expression network analysis. Similar analyses identified three vital modules and 10 hub genes to be associated with fuzz development. The findings in this study contribute to understanding the complex molecular mechanism(s) regulating fibre initiation and development and indicate that *G. arboreum* may have fibre developmental pathways different from tetraploid cotton. It also provides candidate genes for further investigation into modifying fibre development in *G. arboreum*.

## 1. Introduction

Cotton, widely grown in more than 70 countries, is the world’s leading textile crop and an important source of plant oil and protein [1]. There are four cultivated cotton species, including two diploid species, *Gossypium herbaceum* (A_1_) and *G. arboreum* (A_2_), and two tetraploid species, *G. hirsutum* (AD)_1_ and *G. barbadense* (AD)_2_ [2]. Although *G. hirsutum* is the most widely planted species (over 90% of world production), *G. arboreum* remains a significant crop in specific regions of Asia and Africa that are either not environmentally suitable for *G. hirsutum* cultivation or where low input crop management is the norm [3,4,5]. *G. arboreum* is also used in modern tetraploid cotton-breeding programs as a source of germplasm tolerant to multiple abiotic and biotic stresses [3]. Additionally, *G. arboreum*, as a putative donor of the A subgenome of the tetraploid cottons, is a good model species for studies of cotton genomics and genetics, as evaluating molecular mechanisms and physiological factors are simplified in diploid species compared to the tetraploid species [3,6].

Plant trichomes are epidermal outgrowths that have many different physical and physiological functions and vary in structure on different organs [7,8]. Cotton plants vary in the density of trichomes on different vegetative parts, from being highly pubescent to glabrous. It has been shown that cotton species with glabrous leaves and stems, for example, produce cleaner cotton lint and can reduce oviposition by some insect pests [9,10,11,12]. Studies of trichome formation will, therefore, not only aid our understanding epidermal cell differentiation but will also be beneficial to insect resistance breeding. Previous research in *G. hirsutum* has identified five major loci (*T_1_*–*T_5_*) for the leaf/stem trichome trait [13]; among them, *T_1_* was located on chromosome A06 (Chr06) [14]. A homeodomain-leucine zipper gene *HD-1* is thought to be the causative gene for *T_1_* [10,15,16,17,18]. A long terminal repeat retrotransposon insertion in Gb*HD-1_At* results in a lack of pubescence on the stem and leaf epidermis in *G. barbadense* [17,18,19]. The silencing of Gh*HD-1* by RNAi significantly reduced trichome density and delayed the timing of fibre initiation, indicating that the underlying mechanisms regulating trichome and fibre initiation might share some common pathways [15].

Trichomes on the epidermis of cotton ovules develop into two types of fibres, lint fibres that can be easily removed by ginning and fuzz fibres that remain attached after ginning, giving ginned cotton seeds a fuzzy appearance [20,21,22]. Lint fibres initiate on the day of anthesis and can elongate up to about 3.5 cm. Fuzz fibres start initiation at about four days post-anthesis (dpa), with a final length of, at most, 0.5 cm [23]. Seeds without fuzz fibres when ginned appear as black seeds and are commonly called naked seeds [24]. Greater fuzz percentages reduced the ginning rate and increased the power consumption of the ginning process across a range of cotton genotypes [25]. The fuzzless phenotype is thus considered a beneficial ginning trait and is a useful tool to understand the biology, genetics, biochemical, and metabolic processes of cotton fibre development [26,27,28]. There are several naturally occurring and mutagenesis-induced fuzzless accessions in different cotton species. Previous studies have identified multiple loci controlling the fuzzless trait in tetraploid cottons [22,29,30,31]. Two of these have been mapped and investigated in detail. The *N_1_* locus, located on chromosome A12 (Chr12), is a dominant fuzzless locus [32,33]. The *n_2_* locus is recessive that resides on chromosome D12 (Chr26) and is in a position homoeologous to *N_1_* [34]. *MYB25-like* (*MYB-MIXTA-like transcription factor 3*, *MML3*) was identified as the underlying gene of *N_1_* and *n_2_* through map-based cloning and gene expression analysis [22,33]. Recently, *Li_3_* was proposed to regulate lint fibre initiation based on an investigation of the fibreless (fuzzless and lintless) mutant XZ142FLM. *Li_3_* encodes *MYB-MIXTA-like transcription factor4* (*MML4*), a gene located adjacent to *MYB25-like* on chromosome D12 [35].

Based on genetic studies, both dominant and recessive inheritance have been reported for the fuzzless trait in *G*. *arboreum* [6,32,36,37]. A genome-wide association analysis (GWAS) using 158 fuzzy and 57 fuzzless accessions identified one major fuzzless locus located on Chr08 in *G*. *arboreum* [38]. Further genetic mapping using a biparental F_2_ population derived from a cross between a fuzzy (GA0146) and fuzzless (GA0149) accession mapped the dominant fuzzless locus to a 600-kb region containing 10 putative protein-encoding genes, including *GaGIR1* (*Ga08G0121*), a homolog of *AtGIR1* (*GLABRA2*-interacting repressor) of *Arabidopsis thaliana*. Independently, using an F_2_ population derived from a cross between the fuzzless DPL972 and its near isogenic fuzzy line DPL971, Feng et al. [39] fine-mapped the dominant fuzzless gene to a 70-kb region on Chr08 containing seven putative protein-coding genes. Based on RNA-seq and genome sequence comparisons, they speculated that Ga*GIR1* was the likely causative gene [39]. Meanwhile, previous studies in *G. arboreum* have mapped the *fibreless* (lintless-fuzzless) locus *sma-4(ha)*, which co-segregated with the *glabrous* locus to Chr06 [32,37,40,41]. However, the candidate gene for the *sma-4(ha)* locus has not yet been reported.

In the present study, we examined a lintless/glabrous and seven fuzzless *G. arboreum* accessions to investigate the genetics of the lintless and fuzzless traits and to uncover the underlying gene(s) responsible for these traits. Using whole genome re-sequencing and a targeted region(s)-oriented strategy, we established *GaGIR1* and *GaMYB25-like* to be the major candidate genes controlling fuzz development and identified *GaHD-1* to be the candidate gene for the lintless trait in *G. arboreum*. Transcriptome-based weighted gene co-expression network analysis (WGCNA) revealed significant biological processes, gene network modules, and hub genes associated with fibre initiation and development. These findings provide valuable information for elucidating the mechanism of fibre and/or trichome formation and identify key genes as potential targets for improving cotton fibre yield.

## 2. Results

### 2.1. Morphology of the Lintless and Fuzzless Accessions

The phenotype of mature seeds and fibre from the nine parental accessions are shown in Figure 1. The fuzz phenotype after the lint fibres had been pulled from the seeds varied among the seven fuzzless accessions. Accession CHANG ZI #1, NAN.GINGJING, and BANI possessed less fuzz than the others, while accession #47 had the most fuzz amongst the seven fuzzless accessions. All seven fuzzless accessions were more or less tufted (some fuzz fibres attached to the micropylar end of the seeds) and produced normal lint fibres. Accession A2_106 was lintless, although a few shorter lint fibres could occasionally be observed, depending on the environment and position on the plant. The seeds of A2_106 were covered by a sparse layer of short fibres that were shorter and stiffer than the fuzz of BM13H.

The trichome phenotype of the petiole, young leaves, petal, and ovule at 0 dpa from the nine parental accessions was determined and is shown in Figure 2 and Appendix A. No fibre initials were observed on 0 dpa ovules of the lintless accession A2_106, and the fibre initials on 0 dpa ovules from all the seven fuzzless accessions were fewer than those on BM13H. Petioles, young leaves, and petals of A2_106 were all glabrous. Secretory trichomes were still evident on leaves, petioles, stems, and petals of A2_106 (Figure 2), indicating that these are regulated differently to the nonsecretory hair-like trichomes of cotton. The trichome densities on petioles, young leaves, and petals (base region) in the seven fuzzless accessions were lower than that on BM13H but were quite variable in different accessions and on different organs. For example, the petals (base region) from CHANG ZI #1 and BANI had trichomes only on the edges, with few observed on other parts of the petal, while the whole or most of the petal (base region) was covered by trichomes in the other five fuzzless accessions, although less so than on BM13H. The number of fibre initials on 0 dpa ovule of JAI.ZIHUA was the lowest amongst the seven fuzzless accessions, but its trichome density on the petioles, leaves, and petals was greater than CHANG ZI #1 and BANI. These results suggest that fibre initiation on ovule and trichome initiation on other organs may share some common mechanism(s) but are regulated differently, and this may be organ- and/or genetic background-dependent.

### 2.2. Segregation Analysis

The fibre phenotypes of F_3_ seeds (the seed coat being a maternal tissue) from each F_2_ population were scored and used to determine the mode of inheritance of the fuzzless and lintless traits (Table 1). The observed segregation ratio of the lintless trait in the F_2_ population derived from cross H fitted the expected phenotypic segregation ratio of 3:1 (χ^2^ < χ^2^_3:1, 1_ = 3.84), indicating that it was controlled by a single recessive genetic locus, hereafter referred to as *Ga_lintless*. All F_2_ plants producing lintless seeds were also glabrous on their vegetative organs. For the fuzzless trait, more fuzzless segregants were observed in six of the seven F_2_ populations than fuzzy segregants. The only exception was the F_2_ population derived from cross A. The observed segregation ratios of the F_2_ populations derived from crosses D and F fitted the expected phenotypic segregation ratio of 3:1 (χ^2^ < χ^2^_3:1, 1_ = 3.84), suggesting that the fuzzless trait in these two populations is controlled by a single dominant locus. The segregation ratios of the fuzzless trait in the F_2_ populations derived from other crosses, i.e., A, B, C, E, and G, deviated significantly from 3:1, suggesting that the fuzzless trait in these five populations is controlled by more than one genetic locus.

### 2.3. Identification of the Candidate Gene for the Ga_lintless Locus

Previous studies in *G. arboreum* have shown that the recessive glabrous locus orthologous to the *t_1_* locus of tetraploids co-segregated with the recessive fibreless mutation *sma-4(ha)* on Chr06, suggesting that these two loci are either closely linked or are the same genetic locus controlling the two traits [32,37]. Accession A2_106 also showed lintless seeds and glabrous vegetative organs (stem, leaf, and petal) (Figure 1 and Figure 2). To investigate whether *Ga_lintless* co-segregated with the previously identified locus responsible for the *sma-4(ha)* mutation, we developed several KASP markers in the candidate region on Chr06, to which the *sma-4(ha)* mutation was mapped. The results showed co-segregation of the *Ga_lintless* locus with markers single nucleotide polymorphism (SNP)022_024 and SNP025_027 (Figure 3A). Sixteen genes (from *Ga06G1777* to *Ga06G1792*) are annotated in the region defined by the markers SNP022_024 and SNP025_027, including *Ga06G1792* (*GaHD-1*), the ortholog of *G. hirsutum HD-1,* which has been demonstrated to play a role in trichome initiation and development [15]. SNP025_027 was located at the last base of the ninth intron of *GaHD-1*. This KASP marker was designed based on an SNP between BM13H (G) and A2_106 (C) that changed the canonical splicing acceptor (from AG in BM13H to AC in A2_106) and so may affect the splicing of the gene. Between BM13H and A2_106, another synonymous SNP was found at position +4334 bp of *GaHD-1* (Figure 3B).

To determine whether SNP025_027 resulted in a splicing variant of *GaHD-1* in A2_106, the cDNA sequences of *GaHD-1* from young leaves RNA of A2_106 and BM13H were cloned and sequenced. Comparisons between the cDNA sequences and the predicted coding sequence of the wild-type *GaHD-1* revealed a 28-bp deletion of the 10th exon in A2_106, which is predicted to result in a frameshift of the coding sequence and premature stop codon (Appendix A), and it is confirmed by the assemblies of the RNA-seq data. Consequently, the final *GaHD-1* protein sequence of A2_106 is predicted to be 247-aa shorter than that of BM13H; although no known conserved domain was evident in this lost protein region (Appendix A), the truncated protein is likely to be nonfunctional. To investigate whether the changes between A2_106 and BM13H altered the expression of the gene, the expression level of *GaHD-1* in ovules of 0 to 5 dpa from A2_106 and BM13H were compared based on the transcriptome data described in Section 2.6 “Analysis of transcriptomes associated with lint and fuzz initiation”. It was found that the expression levels of *GaHD-1* in A2_106 were significantly decreased at all timepoints (Figure 3C) but were not completely abolished. These results suggest that *GaHD-1* is a likely candidate gene for the *Ga_lintless* locus.

### 2.4. Identification of the Candidate Loci Underlying the Fuzzless Trait

Several studies have reported multiple genes or regions associated with the fuzzless trait in both tetraploid and diploid cotton [22,33,38,39]. To find out whether those reported genes/regions are associated with the fuzzless trait observed in the seven *G. arboreum* accessions used in this study, we designed KASP markers based on SNPs identified between the fuzzless *G. arboreum* accessions and BM13H in those known regions and genotyped individuals from each F_2_ population. Association analysis between genotype and phenotype was then performed. A strong association between the markers on Chr08 and the fuzzless phenotype was found in all the seven F_2_ populations (Table 2). Association, although not as strong as that observed on Chr08, between the markers on Chr12 and the fuzzless phenotype was also found in the F_2_ populations derived from crosses C and G (Table 2). These results suggest the Chr08 (*Ga_Fzl1*) and the Chr12 locus (*Ga_Fzl2*) are the major and minor loci controlling fuzz development in *G. arboreum*, respectively.

To further elucidate the contributions of *Ga_Fzl1* and *Ga_Fzl2* to fuzz development, F_3_ seeds of F_2_ individuals from the cross G (BM13H×JAI.ZIHUA), which were homozygous at the *Ga_ Fzl1* locus but heterozygous at the *Ga_Fzl2* locus, were planted to obtain F_4_ seeds for phenotyping. The fuzz phenotype of seeds with each of the four allele combinations is shown in Figure 4. When *Ga_ Fzl1* was dominant fuzzless homozygous (*Ga_ Fzl1 ^JAI.ZIHUA/ JAI.ZIHUA^*), seeds with homozygous *Ga_Fzl2* from BM13H (*Ga_Fzl2 ^BM13H/BM13H^*) had more fuzz than seeds with homozygous *Ga_Fzl2* from JAI.ZIHUA (*Ga_Fzl2 ^JAI.ZIHUA/JAI.ZIHUA^*). However, when *Ga_ Fzl1* was recessive fuzzy homozygous (*Ga_ Fzl1 ^BM13H/BM13H^*), all seeds showed the fuzzy phenotype regardless of the genotype of the *Ga_Fzl2* locus. This result indicates that *Ga_Fzl1* contributes more than *Ga_Fzl2* to fuzz development and suggests that *Ga_ Fzl1* is in recessive epistasis to *Ga_Fzl2*.

### 2.5. Potential Candidate Genes for the Ga_Fzl1 and Ga_Fzl2 Loci

The *Ga_Fzl1* locus contains *GaGIR1* (*Ga08G0121*), and the *Ga_Fzl2* locus contains *GaMYB25-like* (*Ga12G1199*). These genes have been previously reported to be involved in fuzz development in *G. arboreum* [39] and the tetraploids [22,33], respectively. We then compared sequences (coding sequence plus ~2000-bp upstream promoter sequence) of these two candidate genes between the different fuzzless *G. arboreum* accessions and BM13H. For *GaGIR1*, three haplotypes were identified amongst the eight *G. arboreum* accessions. BM13H belongs to haplotype I. The fuzzless accessions #47, NANTONG, JAI.ZIHUA, #6, and NAN.GINGJING belong to haplotype Ⅱ, which showed multiple sequence variations compared to BM13H, including a 3-bp deletion and a nonsynonymous SNP (amino acid changed from proline to alanine) in the coding region and a 24-bp (ATAT repeats) deletion and 12 SNPs in the promoter region. The other two fuzzless accessions, CHANG ZI #1 and BANI, belong to Haplotype Ⅲ, which have coding sequences identical to that of BM13H but with a 4-bp deletion in the promoter region instead of the 24-bp deletion observed in haplotype Ⅱ accessions (Figure 5A). For *GaMYB25-like*, compared to BM13H, the fuzzless accessions JAI.ZIHUA and #6 possessed nonsynonymous SNPs, which caused an amino acid change from alanine to serine. While the promoter regions of these two accessions, and both promoter and coding regions of the other fuzzless accessions, were identical to those of BM13H (Figure 5B).

To obtain further information about these two candidate genes, their expression levels in developing ovules from 0 to 5 dpa were analysed from RNA-seq data in BM13H, JAI.ZIHUA, and CHANG ZI #1, representing the different haplotypes, as described in Section 2.6 “Analysis of transcriptomes associated with lint and fuzz initiation”. The expression levels of *GaGIR1* were consistently significantly higher in the two fuzzless accessions than in BM13H (Figure 5C), suggesting that *GaGIR1* might be acting as a repressor of fuzz initiation. The expression levels of *GaMYB25-like* in 0 to 3 dpa ovules were lower in JAI.ZIHUA than in BM13H, with a significant decrease observed in 0 to 3 dpa ovules (Figure 5D). These results provide further support for *GaGIR1* and *GaMYB25-like* being the candidate genes for the *Ga_Fzl1* and *Ga_Fzl2* loci, respectively.

### 2.6. Analysis of Transcriptomes Associated with Lint and Fuzz Initiation

To globally identify the differentially expressed genes (DEGs) between BM13H and a subset of the fibre mutants, and to explore the mechanism(s) regulating fuzz and lint initiation, RNA-seq analysis was performed using 0, 1, 3, and 5 dpa whole ovules from the fuzzless accessions JAI.ZIHUA (FL1) and CHANG ZI #1 (FL2), the lintless accession A2_106 (LL), and the normal fuzzy-linted accession BM13H (Normal). Approximately 1092 million clean paired-end reads were obtained from a total of 48 samples, and the average Q30 percentage (sequences with sequencing error rates lower than 0.1%) was 94.85%, indicating high quality of the RNA-seq data. Clean reads (79.1–83.8%) from these samples were aligned to the *G. arboreum* reference genome [38], of which, about 3.5–7.5% were multiple mapped reads (Appendix A). Cluster analysis based on the expression pattern and level of each transcript from all samples was carried out to identify the transcriptome similarities between samples (Appendix A). The result showed that the transcriptomes of 3 dpa and 5 dpa samples from FL1, FL2, and Normal clustered together, while the transcriptome of 3 dpa and 5 dpa samples from LL clustered with the transcriptomes of 0 dpa and 1 dpa samples from all four accessions, probably because no lint initials are produced at these stages on the ovules of LL.

Figure 6A shows the number of DEGs in LL vs. Normal at different timepoints. Overall, there were 1525 nonredundant upregulated and 2295 nonredundant down-regulated DEGs (Appendix A), and the number of DEGs, particularly those that were down-regulated, increased with the progression of ovule development. In total, there were 155 (10.16%) upregulated and 408 (17.78%) down-regulated DEGs identified in all four timepoints (Figure 6B,C). The clustering heatmap of the common DEGs demonstrated a clear separation of the LL and Normal groups (Appendix A).

To identify possible biological functions that are modulated by the LL mutant gene(s), gene ontology (GO) enrichment of all the nonredundant upregulated and down-regulated LL-associated DEGs was performed. Several of the top GO terms in the upregulated and down-regulated groups were shown in Figure 7A, B, respectively, and the details of more enriched categories are summarised in Appendix A and Appendix A. The 1525 upregulated LL DEGs were enriched in 11 biological process GO terms, 6 cellular components GO terms, and 22 molecular function GO terms. Among them, cell wall biogenesis, apoplast, and oxidoreductase activity were the most enriched GO terms in each GO class. Twenty-two biological processes, including lipid biosynthetic process and xyloglucan metabolic process; 4 cellular components, such as microtubule; and 22 molecular function GO items, including ADP-binding and transferase activity, were significantly depressed in LL. These results reveal the complexity of lint fibre initiation and elongation and highlight the important biological functions of the DEGs in cotton fibre development.

The DEGs between the fuzzless accessions (FL1 and FL2) and Normal genotype were also analysed. The distribution of DEG numbers at different timepoints are shown in Figure 8A. The number of FL DEGs was higher in 5 dpa ovules than at the other timepoints, consistent with fuzz being initiated and developed later than lint at around 3–5 dpa. In total, there were 721 upregulated and 1663 down-regulated DEGs in FL1 vs. Normal, while 263 FL DEGs were common in all timepoints. In the comparison of FL2 vs. Normal, 2113 DEGs were identified, including 601 upregulated and 1512 down-regulated DEGs, 211 of which were upregulated or down-regulated in all four timepoints. We also compared the number of DEGs from FL1 vs. Normal and FL2 vs. Normal, at each timepoint, and found 190, 185, 217, and 711 common FL DEGs in both comparisons at 0, 1, 3, and 5 dpa, respectively (Figure 8B and Appendix A).

As the fuzzless traits of both JAI.ZIHUA and CHANG ZI #1 seemed to be controlled by *Ga_Fzl1* (*GaGIR1*), the FL DEGs (250 nonredundant upregulated DEGs and 633 nonredundant down-regulated DEGs) found in both FL1(JAI.ZIHUA) vs. Normal and FL2 (CHANG ZI #1) vs. Normal comparisons are likely to function downstream of *Ga_Fzl1* in regulating fuzz development. They were thus further analysed for GO enrichment. In total, 13 molecular function GO terms, including sequence-specific DNA-binding, oxidoreductase activity, and transporter activity; five biological process GO terms, including transport and cellulose biosynthetic process; and the cellular component GO term tRNA (m1A) methyltransferase complex, were found to be enriched in FL analyses (Figure 8C and Appendix A).

### 2.7. Co-Expression Network Analysis of Genes Associated with Lint Initiation

To gain further insight into the gene regulatory network and gene modules associated with lint fibre initiation, WGCNA was applied to establish co-expression networks for a total of 1639 nonredundant DEGs that were identified in pairwise comparisons between the fibre mutants (LL, FL1, and FL2) and Normal in 0 dpa ovules. Eight distinct modules were identified (Appendix A). Based on the number of lint fibre initials present on the surface of 0 dpa ovules (Figure 2D), the lint fibre initiation trait was scored as 0, 1, 2, and 3 in LL, FL1, FL2, and Normal, respectively. Module trait-associated analysis uncovered significant association of three modules (MEblue, MEbrown, and MEturquoise) with lint fibre initiation (Appendix A). These three modules contained 339, 293, and 501 DEGs, respectively (Appendix A). The module membership and gene significance were highly correlated in each module (Appendix A).

Eight, two, and sixteen hub genes were identified in the modules MEblue, MEbrown, and MEturquoise, respectively. The expression levels of these hub genes in 0 dpa ovules from the four *G. arboreum* accessions and their descriptions were shown in Figure 9A. All eight hub genes in the module MEblue were significantly down-regulated in LL, FL1, and FL2; the two hub genes in the module MEbrown were correlated with the density of fibre initials; while the expression levels of the sixteen hub genes in module MEturquoise were significantly upregulated only in LL. Based on GO enrichment analysis, module MEbrown mainly enriched in biological processes such as the fatty acid biosynthetic process and lipid biosynthetic process; module MEblue mainly enriched in biological processes including response to auxin, and molecular functions including ADP-binding and oxidoreductase activity; while module MEturquoise mainly enriched in ADP-binding and oxidoreductase activity (Appendix A). These results revealed the complicated gene networks involved in cotton lint fibre formation and provided key genes for further study to understand the molecular mechanism(s) associated with lint fibre initiation and development.

### 2.8. Co-Expression Network Analysis of Genes Associated with Fuzz Development

WGCNA was also performed to understand the gene regulatory networks and modules associated with fuzz development. A total of 4371 nonredundant DEGs observed in 5 dpa ovules were used in the analysis. Ten modules were detected (Appendix A). According to the amount of fuzz produced on mature seeds, the fuzz development trait was scored as 1, 0, 0, and 2 in LL, FL1, FL2, and Normal, respectively. Three modules, including MEgreen, MEred, and MEblack, were identified to be highly associated with fuzz development (Appendix A). These three modules contained 244, 228, and 204 DEGs, respectively (Appendix A). Eight and two hub genes were identified in modules MEgreen and MEred, respectively, while no hub genes meet the threshold of K_ME_ > 0.99 in module MEblack. Two hub genes of module MEred were down-regulated in the two fuzzless accessions and the lintless accession. Eight hub genes of module MEgreen had expression levels higher in the fibre mutants than in BM13H, particularly in FL1 (JAI.ZIHUA) (Figure 9B). The module membership and gene significance were highly correlated in each module (Appendix A).

GO enrichment analysis indicated that module MEblack was mainly enriched in molecular function ADP-binding, DEGs in module MEred mainly participated in biological processes intracellular signal transduction, and DEGs in modules MEgreen were enriched in terms related to microtubules (Appendix A). These findings again provide genes and biological functions correlated with fuzz development and candidate genes for further studies.

## 3. Discussion

The molecular mechanism governing fibre and trichome initiation in cotton is complicated and involves many different biochemical and physiological pathways. Understanding the fibre development will be crucial for breeding high lint yield and high-quality cotton varieties. Several genes have been identified and verified to be functional in fibre initiation in *G*. *hirsutum* and *G. barbadense* [31]. However, little is currently known about fibre and trichome initiation in *G. arboreum. HD-1* has been well-studied in both *G. hirsutum* and *G. barbadense*. Partial silencing of both homoeologs of *HD-1* in tetraploid *G. hirsutum* resulted in a glabrous phenotype and delayed the timing of fibre initiation [15]. However, the hairless stem and leaf phenotype observed in *G. barbadense* is believed to be caused by a retrotransposon insertion in *GbHD-1_At* [17,18], making it nonfunctional, and *GbHD-1_Dt* might only be responsible for trichomes on the lower side of leaves and new buds [18]. In this study, we identified *GaHD-1* as a potential candidate gene of the *Ga_lintless* locus responsible for both the glabrous and lintless phenotypes of the *G. arboreum* accession A2_106. An SNP mutation in the canonical splicing acceptor in A2_106 caused alternative splicing that led to a loss of 247 aa in *GaHD-1*. It is not known whether the altered protein retains any function, although the recessive nature of the mutation would suggest that it does not. This change to *GaHD-1* is the most likely genetic basis for the mutant phenotypes observed in A2_106. Loss-of-function or reduced expression of *HD-1* in both tetraploid species impaired trichome initiation and development in vegetative organs but only delayed fibre initiation on the seed coat (at least in *G. hirsutum*) and had no impact on lint development in *G. barbadense*. Possible explanations for the observed difference between the *G. arboruem* and tetraploid function of *HD-1* is that there is some low threshold in the expression of *GhHD-1* (either from both homoeologs in *G. hirsutum* or just the D subgenome in *G. barbadense*) in the seed coat epidermis required for lint fibre initiation and development or, possibly, that the seed fibre pathways may be regulated differently between the diploid and tetraploid cottons, with the tetraploid species having unknown genetic component(s) that can compensate for the loss-of-function of *HD-1*.

*GaGIR1* was first identified as the candidate gene responsible for the fuzzless trait in *G. arboreum* accession DPL972 based on its promoter and coding sequence variation and significantly higher expression levels in the mutant than the wild-type [39]. We found that *GaGIR1* was also associated with the fuzzless phenotype observed in all seven of the fuzzless *G. arboreum* accessions used in this study, which could be classified into two *GaGIR1* haplotypes (Ⅱ and Ⅲ) based on the sequence variations found in their promoter and coding regions (Figure 5A). The only difference between haplotype Ⅲ fuzzless accessions and BM13H (normal fuzzy seeds) was a 4-bp deletion in the promoter (~260-bp upstream of the start codon), suggesting a potential role for the timing or expression level of *GaGIR1* in fuzz initiation. Between haplotype Ⅱ fuzzless accessions and BM13H, sequence variations were observed in both the promoter and coding region of *GaGIR1* (Figure 5A), including a 3-bp (TTG) deletion. These are the same three nucleotides missing from the fuzzy parent (DPL971) of DPL972, so may just represent a common allele of *Ga_GIR1* that we also see in *GhGIR1_At* (*GH_A08G0106*) and *GbGIR1_At* (*GB_A08G0101*) (Appendix A), suggesting that it may not contribute to the fuzzless trait. Haplotype Ⅱ fuzzless accessions also had a nonsynonymous SNP in the coding region downstream from the TTG deletion that changed a proline to an alanine. This SNP was not present in the fuzzless DPL972 accession and, so, represents another distinct haplotype for *GaGIR1* [39]. It is unknown whether either of these coding sequence changes would affect protein function, but as they do not occur within any known functional domains, the expectation is they are unlikely to contribute to the fuzzless trait. A 4-bp and 24-bp deletion in the core promoter of *GaGIR1* was observed in both haplotype Ⅲ and Ⅱ of the fuzzless accessions, respectively, but no such deletions were reported in the promoter of DPL972 [39]. However, haplotype Ⅱ accessions and DPL972 shared two common SNPs at positions -500-bp and -695-bp (Figure 5A) that were not present in haplotype Ⅲ. These results suggest that both the deletion, particularly the 4-bp deletion observed in haplotype Ⅲ fuzzless accessions, and the two common SNPs observed in DPL972 and haplotype Ⅱ accessions, are important for the large increases in expression of *GaGIR1* and, hence, loss of fuzz development. Together, these results suggest that the sequence variations in the promoter of *GaGIR1,* rather than those in the coding region, are the underlying reasons for the fuzzless phenotype. In *Arabidopsis*, *GIR1* has been reported to be a repressor of *GL2*, and the loss-of-function mutant of *GIR1* showed an enhanced development of root hairs [42]. In this study, we observed that the expression level of *GaHD-1*, a similar transcription factor to the *Arabidopsis GL2,* and important in trichome and lint development*,* was decreased in the two fuzzless accessions, probably due to the significant upregulation of *GaGIR1* (Appendix A), suggesting that *GaGIR1* may also repress the expression level of *GaHD-1* in cotton.

In the GWAS study by Du et al. [38], 158 fuzzy and 57 fuzzless *G. arboreum* accessions (including CHANG ZI #1) were used to identify the region on Chr08, containing ten genes that included *GaGIR1* (*Ga08G0121*), to be associated with the fuzzless trait, although they suggested *Ga08G0117* (a casparian-strip membrane protein) as a potential candidate gene. In this study, we confirmed this region and identified another region containing *GaMYB25-like* (*Ga12G1199*) on Chr12 that was also associated with the fuzzless trait, although its effect on fuzz development appears to be weaker. The association between *GaMYB25-like* and the fuzzless trait was only detected in two of the seven fuzzless accessions (JAI.ZIHUA and #6). In addition, *GaGIR1* was found to be in recessive epistasis to *GaMYB25-like* in *G. arboreum*. These results suggest that the region containing *GaGIR1* is the major locus underlying fuzz development in *G. arboreum*, while the region containing *GaMYB25-like* is a minor locus. This is different from what has been reported in tetraploid cottons, where *MYB25-like* was identified as the causative gene for the two major fuzzless loci, *N_1_* and *n_2_* [22,33], and *GIR1* has not yet been identified to be associated with fuzz development in the tetraploid cottons. The different effects of *MYB25-like* on fuzz development in tetraploid cottons and diploid cottons may be a result of different genetic backgrounds or the nature of the specific mutation(s), as the expression level of *MYB25-like* homoeologs has recently been suggested to play a significant role in fuzz initiation [22]. *GaMYB25-like* may not have been identified by the GWAS study [38] because of the limited diversity of the accessions used, or this locus was filtered out by minor allele frequency (MAF) cut-offs. These results suggest that simple genetic crosses, as well as GWAS, are needed to uncover all the genetic components underlying complex agronomic traits. It has been reported, for example, that there are at least five distinct loci associated with the fuzzless trait in *G. barbadense* [22]. In *G. arboreum*, besides the two loci mentioned above, there are likely to be other yet-to-be-characterised loci that contribute to fuzz development based on the distorted segregation ratios of fuzzy and fuzzless seeds in a number of the F_2_ populations used in this study. The cross A (BM13H × #47), for example, had more fuzzy offspring than fuzzless offspring, even though the dominant fuzzless locus *Ga_Fzl1* was identified to be present in this population.

Previous reports have identified many genes involved in cotton fibre initiation, such as genes of the MYB family [33,35,43,44,45] and genes involved in hormone biosynthesis pathways [46,47]. RNA-seq analysis has also been used to identify many biological functions and gene networks involved in fibre initiation over the past several years [48,49,50]. However, most research has focused on *G. hirsutum*. In this study, we used *G. arboreum*. Apart from the identification of important biological processes or pathways, we identified three modules with 26 hub genes to be highly associated with lint fibre initiation and three modules with 10 hub genes to be highly correlated with fuzz development based on WGCNA. Only two genes, *Ga11G0310* (kinesin-like protein) and *Ga14G2148* (uncharacterised protein) were identified as hub genes with significant associations with both lint fibre initiation and fuzz development, suggesting the two types of fibres may have very different developmental programs. Some biological functions that have been previously identified to be related to fibre initiation and development, such as fatty acid and lipid biosynthetic process and microtubule-based movement, were also enriched in the *G. arboreum* fibre and fuzz networks; however, several hub genes found in these modules are uncharacterised, such as *Ga11G3722*, *Ga07G1739,* and *Ga14G2148*. Further investigation of these hub genes will be helpful in understanding the molecular mechanism(s) controlling cotton fibre initiation and development in this cotton species.

## 4. Materials and Methods 

### 4.1. Plant Materials

Seven linted-fuzzless, one lintless-fuzzy, and glabrous (A2_106) and one normal (linted-fuzzy) fibre (BM13H) *G. arboreum* accessions were used in this study (Table 1 and Appendix A). The phenotypic characteristics of BM13H have been previously described [51]. The eight fibre mutant accessions were obtained from the cotton collection of the Crop Germplasm Research Unit of the United States Department of Agriculture, Agriculture Research Service (USDA-ARS) located in College Station, Texas, USA. All eight fibre mutant accessions were crossed with BM13H to establish F_2_-segregating populations for genetic analysis (Table 1). Plants were grown in glasshouses at about 28 ± 2 °C with natural lighting in Canberra, Australia.

### 4.2. DNA Sample Preparation, Sequencing, and SNP Genotyping

The genomic DNA of *G. arboreum* accessions and their F_2_ individuals were extracted from young leaves according to the method described by Ellis et al. [52]. DNA libraries for whole genome re-sequencing were created for each accession and sequenced in an Illumina HiSeq platform according to the manufacturer’s instructions by the Beijing Genomics Institute (BGI) (Beijing, China). Approximately 100 Gb of 150-bp paired-end reads were generated for each accession. Raw reads were first checked for quality by FastQC v0.11.8 [53] and were then mapped to the *G. arboreum* reference genome [38] by BWA v0.7.17 [54]. Single nucleotide polymorphisms (SNPs) were called by SAMtools v1.4 [55] and BCFtools v1.9.0 [56]and visualised by IGV v2.4.14 [57].

Kompetitive allele specific PCR (KASP) markers were designed based on SNPs between parental accessions and were used to detect the genotype of F_2_ individuals. KASP genotyping was performed as previously described [58]. All the oligonucleotides used in this study are listed in Appendix A.

### 4.3. Phenotyping

The phenotypes of fuzzless and lintless seeds were scored based on visual inspection. Seeds of F_2_ populations from fuzzless lines were scored as fuzzy and fuzzless, and those from the lintless line were scored as normal and lintless-fuzzless.

Scanning electron microscopy (SEM) images of ovules and petals (base region) at 0 dpa, young leaves, and petioles were taken with Zeiss EVO LS 15 environmental SEM (Jena, Germany). More than ten ovules, two duplicates of young leaves, petioles, and petals from each sample were used for analysis. SEM samples were prepared as described previously [7,59].

### 4.4. Association and Linkage Analyses

Fisher’s exact test was performed to determine an association between markers and traits. A marker was considered to be significantly associated with a trait when the *p*-value was < 0.05. Linkage analysis and genetic map construction were performed using the software JoinMap 4.0 [60].

### 4.5. Total RNA Extraction and Transcriptome Sequencing

Total RNA samples of ovules from A2_106, JAI.ZIHUA, CHANG ZI #1, and BM13H at 0, 1, 3, and 5 dpa were extracted using the Maxwell RSC Plant RNA kit with the Maxwell RSC Instrument (Promega). Each sample included three biological replicates. The quality of the RNA samples was checked using the Agilent 2100 Bioanalyzer System (Agilent Technologies, Palo Alto, CA, USA). RNA samples from each timepoint with an RNA integrity number (RIN) above 7.0 were used for RNA-seq library construction. Libraries were sequenced using the paired-end configuration on an Illumina HiSeq instrument by BGI according to the manufacturer’s instructions (Illumina, San Diego, CA, USA).

### 4.6. RNA-seq Data Analysis

Raw reads were first processed using Trimmomatic v0.39 [61] to remove low-quality sequences and adaptors. The quality of trimmed FASTQ files was evaluated using FastQC v0.11.8 [53]. The clean reads were mapped to the reference *G. arboreum* genome [38] using Biokanga [62], and transcript per million mapped reads (TPM) was calculated for estimating gene expression levels with a custom python script. HTSeqcount [63] was used to get the raw counts of each gene. The function “hclust” in R was used to produce the hierarchical clustering analysis of samples. The R package DESeq2 was used to identify differentially expressed genes (DEGs) [64]. The adjusted *p*-value < 0.05 and an absolute value of log2 fold change > 1 were used as criteria for the determination of DEGs. The raw RNA-Seq data is available from https://doi.org/10.25919/5e13cb617475a.

### 4.7. Gene Ontology and Pathway Analysis

Gene ontology (GO) enrichment analysis was performed to determine the main biological functions of the DEGs. The R package ClusterProfiler [65] was used to perform GO enrichment analysis with a *p*-value < 0.05 and minimum gene number ≥ 4 as the cut-off.

### 4.8. Construction of Co-Expression Network and Visualisation

Co-expression networks were constructed by the R-based package WGCNA [66]. Nonredundant DEGs at 0 and 5 dpa in A2_106 vs. BM13H, JAI.ZIHUA vs. BM13H, and CHANG ZI #1 vs. BM13H were used to construct co-expression networks for lint fibre initiation and fuzz development. The modules were identified using the automatic one-step network construction approach with the minModuleSize being set to 30. Modules that significantly associated with traits were identified with a *p*-value < 0.01 and a correlation coefficient > 0.7, and hub genes in these modules were selected based on a module membership (K_ME_) > 0.99.

## 5. Conclusions

In this study, a recessive locus containing *GaHD-1* on Chr06 was identified as the likely gene to control the lintless trait in one accession of *G. arboreum*, while the region containing *GaGIR1* and the region containing *GaMYB25-like* were shown to be the major and minor loci associated with fuzz development in this species. Analyses of transcriptome and gene co-expression networks revealed important biological processes, gene modules, and hub genes controlling initiation and development of *G. arboreum* fibres. Our results provide a solid foundation for further understanding the molecular mechanism of fibre formation and candidate genes for improving fibre yield through transgenic approaches.

## Figures and Tables

**Figure 1 ijms-21-01675-f001:**
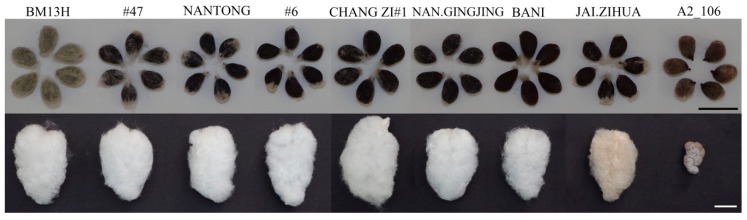
Seed and fibre phenotype of *G. arboreum* accessions used in this study. Lower panel shows a locule of seeds as they are found in the open boll, and the upper panel is the individual seeds with the lint fibres removed by hand. Accession JAI.ZIHUA had light brown lint fibres (scale bar = 1cm).

**Figure 2 ijms-21-01675-f002:**
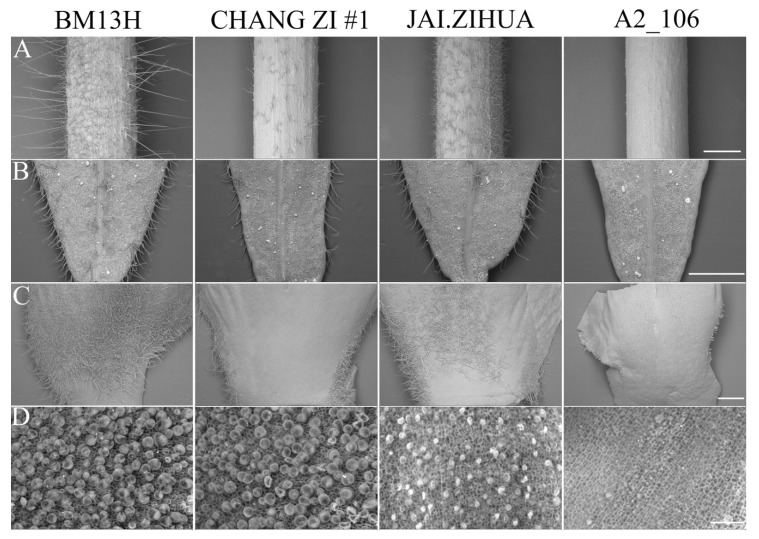
Comparison of trichome phenotype of different tissues from fuzzless (CHANG ZI #1 and JAI.ZIHUA), lintless (A2_106), and normal fibre (BM13H) *G. arboreum* accessions. (**A**) Trichome phenotype of young petioles (scale bar = 1 mm), (**B**) trichome phenotype of young leaves (scale bar = 1 mm), (**C**) trichome phenotype of petals near the base (scale bar = 1 mm), and (**D**) fibre initials on 0 dpa ovules (Scale bar = 0.05 mm).

**Figure 3 ijms-21-01675-f003:**
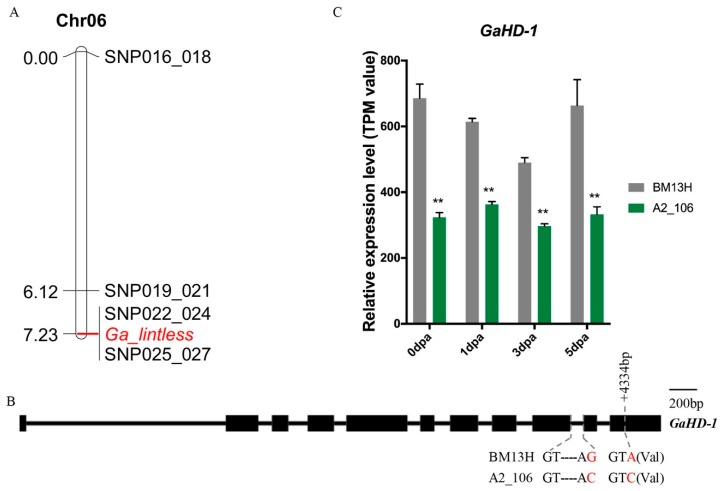
Genetic mapping of *Ga_lintless* and comparison of sequence and expression level of *Ga_HD-1* between A2_106 and BM13H. (**A**) Genetic linkage map of *Ga_lintless* on Chr06 with distances marked in centimorgans, and (**B**) gene structure and sequence variations of *Ga_HD-1* between BM13H and A2_106. Exons are indicated by the black boxes. Variation in nucleotide sequences between the two accessions are indicated in red font. (**C**) Relative expression level of *GaHD-1* in A2_106 and BM13H. Values are in transcripts per million mapped reads. ** indicates the differences relative to BM13H were significant, with a probability of 0.01.

**Figure 4 ijms-21-01675-f004:**
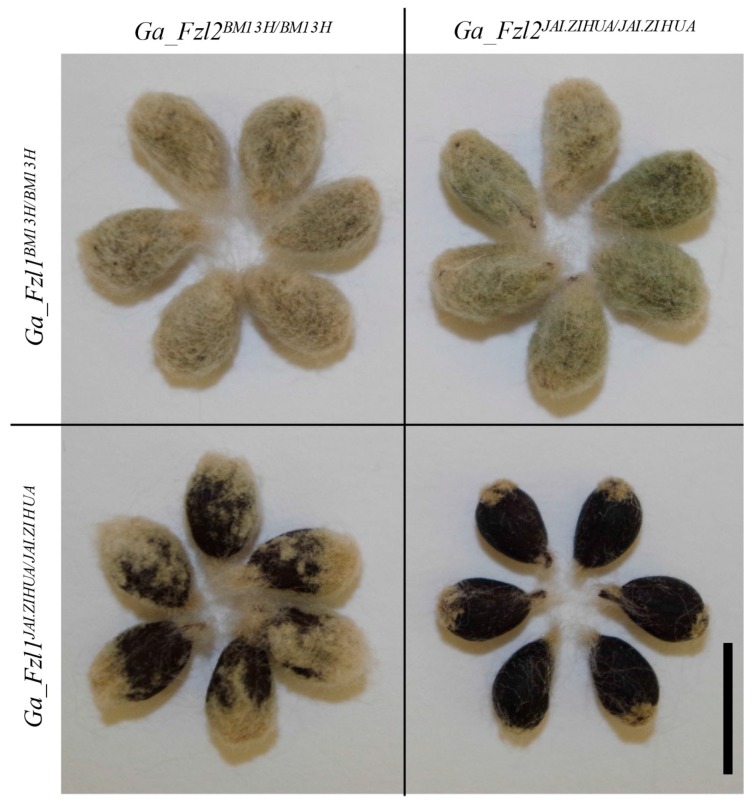
Phenotype segregation analysis of seeds with different combinations of *Ga_Fzl1* and *Ga_Fzl2* alleles (scale bar = 1 cm). Row names indicate the genotype of *Ga_Fzl1*, and column names indicate the genotype of *Ga_Fzl2*.

**Figure 5 ijms-21-01675-f005:**
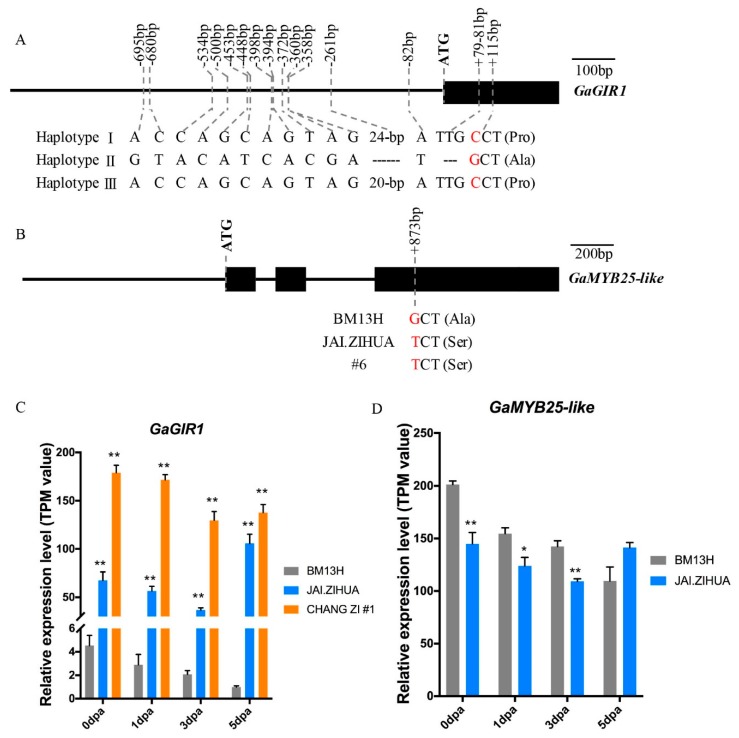
Comparison of *GaGIR1* and *GaMYB25-like* between BM13H and the fuzzless accessions. (**A**) Sequence comparison of the promoter and ORF regions of *GaGIR1* in different accessions. Only nucleotides different to BM13H are shown and SNPs that result in amino acid changes are in red. Haplotype I is BM13H. Haplotype Ⅱ contains accessions #47, NANTONG, #6, NAN.GINGJING, and JAI.ZIHUA. Haplotype Ⅲ contains accessions CHANG ZI #1 and BANI. (**B**) Sequence comparison of the promoter and ORF regions of *GaMYB25-like*. Exons are indicated by the black boxes, and SNPs that result in amino acid changes are in red. (**C**) Expression profiles of *GaGIR1* in 0-5 dpa ovules of the fuzzless accessions JAI.ZIHUA, CHANG ZI #1, and fuzzy-linted accession BM13H. (**D**) Expression profiles of *GaMYB25-like* in 0-5 dpa ovules of the fuzzless accession JAI.ZIHUA and fuzzy accession BM13H. * and ** indicate the significant differences with a probability of 0.05 and 0.01, respectively.

**Figure 6 ijms-21-01675-f006:**
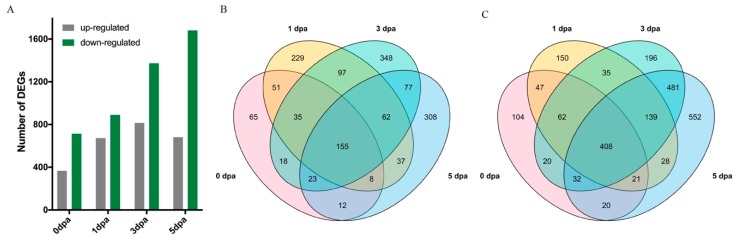
Transcriptome analysis of the lintless accession. (**A**) Number of differentially expressed genes (DEGs) identified in pairwise comparisons between lintless accession A2_106 (LL) and normal fuzzy-linted accession BM13H (Normal) ovules. (**B**) Venn diagram showing upregulated DEGs in LL relative to Normal at each of the four timepoints. (**C**) Venn diagram showing down-regulated DEGs in LL relative to Normal. Numbers of overlapping genes between timepoints are indicated.

**Figure 7 ijms-21-01675-f007:**
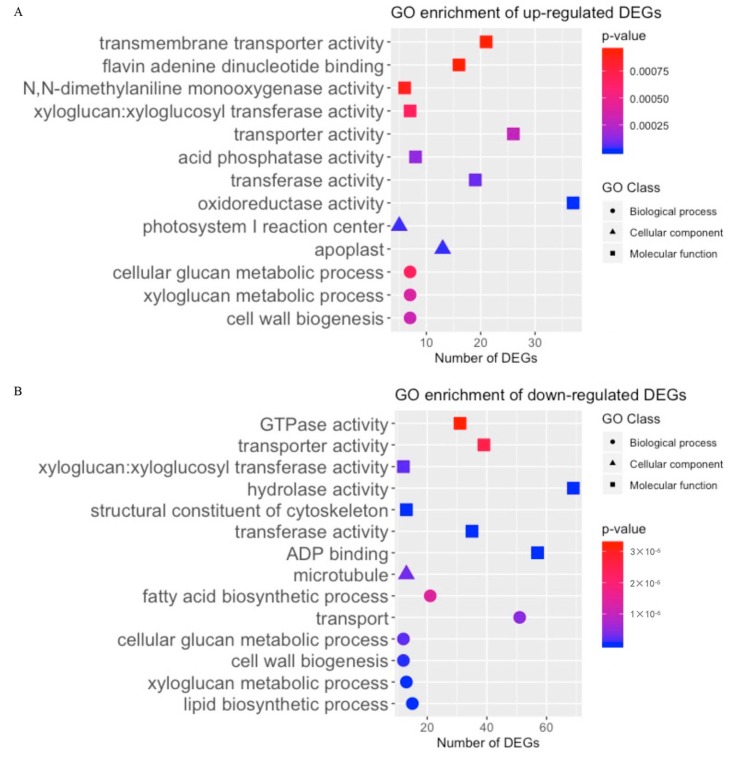
Gene ontology (GO) enrichment results of nonredundant DEGs in the lintless accession. (**A**) GO enrichment results of nonredundant upregulated DEGs in the LL group. (**B**) GO enrichment results of nonredundant down-regulated DEGs in the LL group.

**Figure 8 ijms-21-01675-f008:**
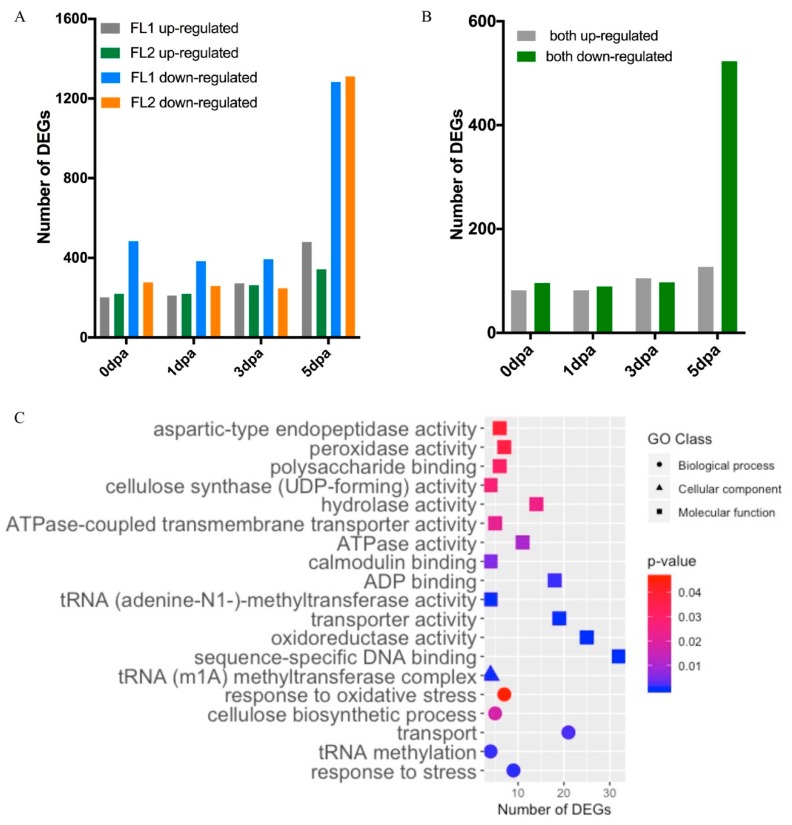
Transcriptome analysis results of the fuzzless accessions (FL). (**A**) Number of DEGs identified in pairwise comparison between FL1 and Normal and FL2 and Normal. (**B**) Number of DEGs that were up- or down-regulated in both FL1 and FL2. (**C**) Enriched GO terms based on DEGs up- and down-regulated in both FL1 and FL2.

**Figure 9 ijms-21-01675-f009:**
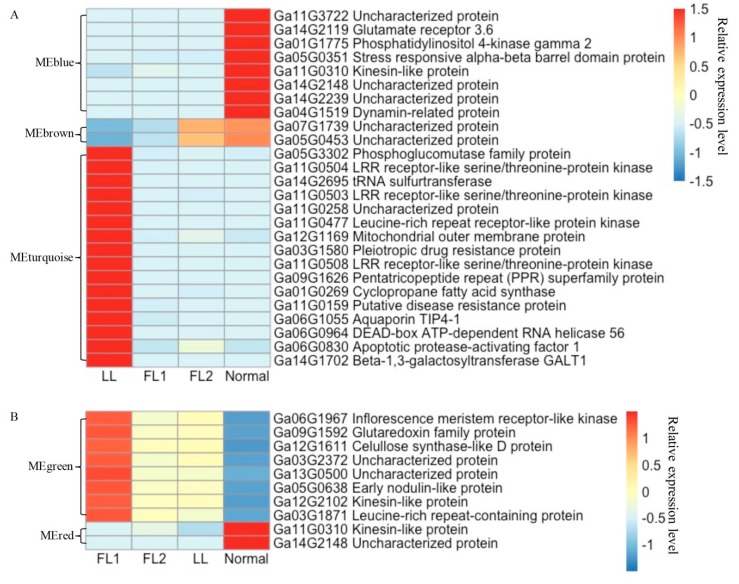
Relative expression heatmap of hub genes detected by weighted gene co-expression analysis (WGCNA). (**A**) Expression heatmap and description of hubs genes associated with lint fibre initiation. (**B**) Expression heatmap and description of hubs genes associated with fuzz development.

**Table 1 ijms-21-01675-t001:** Genetic analysis of the fuzzless and lintless traits in segregating populations.

Cross ID	Parents		Phenotype			Phenotype		
	Female	Male	Fuzzless	Fuzzy	χ^2^	Normal	Lintless	χ^2^
A	BM13H	#47	28	43	47.89			
B	BM13H	NANTONG	49	28	5.30			
C	BM13H	#6	44	32	11.86			
D	BM13H	CHANG ZI #1	63	27	1.20			
E	BM13H	NAN.GINGJING	59	31	4.28			
F	BM13H	BANI	57	25	1.32			
G	BM13H	JAI.ZIHUA	51	42	20.16			
H	BM13H	A2_106				67	23	0.01

**Table 2 ijms-21-01675-t002:** Association analysis between markers and the fuzzless trait. SNP: single nucleotide polymorphism.

Cross ID	Marker	Chromosome	Physical Position	*p*-Value
A	SNP004_006	Chr08	905,375	< 0.0001
B	SNP004_006	Chr08	905,375	< 0.0001
C	SNP004_006	Chr08	905,375	< 0.0001
C	SNP001_003	Chr12	14,892,043	0.0186
D	SNP010_012	Chr08	906,144	< 0.0001
E	SNP004_006	Chr08	905,375	< 0.0001
F	SNP010_012	Chr08	906,144	< 0.0001
G	SNP004_006	Chr08	905,375	< 0.0001
G	SNP001_003	Chr12	14,892,043	< 0.0001

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
