# Peer review of "Genetic Identification and Transcriptome Analysis of Lintless and Fuzzless Traits in Gossypium arboreum L."

_ijms, 2020, doi:10.3390/ijms21051675_

Round 1

Reviewer 1 Report

This MS used genetic and association analysis to identify a gene that potentially regulates the lintless trait in G. arboreum. They authors also performed transcriptome analysis using ovules from cultivars of contrasting phenotypes and report on the presence of genes differentially regulated among the different accessions. It is clear that the manuscript provides a significant amount of information to the field but I have some concerns that need to be addressed.  

My major concern on the way the result of transcriptome analysis using RNA-seq is presented. The authors report mainly on the number of DEG and the number of functional categories enriched in the analysis. The number of upregulated and downregulated genes were reported in different groups.

  1. Specific genes/gene families that are differentially regulated in the different accessions are not mentioned in the results. 
  2. It is not clear which of the genes that are differentially regulated are actually associated with lintless or fuzziness trait.
  3. How was the expression of genes such as GaHD1 in the RNA-seq analysis
  4. Are the genes differentially regulated based on the RNA-seq data are validated using quantitative PCR?
  5. The conclusion that GaHD-1 is a candidate gene responsible for the Ga-lintless locus is not validated by functional studies, and needs to be tonned-down.  For example, Line 178-9. Fig. 3C. shows a clear decrease in relative expression, but it is not completely suppressed in A2-106. This difference may not be translated into expression at protein level.   
  6. Specific genes/gene families identified in the RNA-seq are not discussed

Author Response

Specific genes/gene families that are differentially regulated in the different accessions are not mentioned in the results. 

The RNAseq data generated a large list of genes that were differentially expressed, 3820 non-redundant genes between the Lintless Accession and Normal, and 883 between Fuzzless accessions and normal). Most of the genes listed are uncharacterised in G. arboreum and so we felt it was better to investigate and detail biological processes and gene modules that were altered rather than focusing on specific genes. All genes that were found to be differentially expressed are listed in the Supplementary Tables S4 and S7, so readers interested in a specific gene can find whether it was differentially expressed. Genes found associated with specific biological processes and significant gene modules are also listed in the supplementary data Tables. The only exceptions to this was genes that were characterised and potentially candidate genes for the traits we were investigating, eg., GaGIR1, GaMYB25-like and GaHD-1 for which we presented their expression profiles in the manuscript. Given the lack of characterised genes involved in G. arboreum lint fibre and fuzz formation, we believe the way we have presented the data is the right balance.

It is not clear which of the genes that are differentially regulated are actually associated with lintless or fuzziness trait.

The RNAseq data is complicated as it involves multiple timepoints and multiple accessions. We have endeavoured to make this clear throughput the manuscript by abbreviating Lintless associated analyses LL, and Fuzzless associated analyses, FL.  All Figures and Tables indicate whether the data involves LL or FL analyses. The experiment is detailed at the start of section 2.6 and the material and methods section 4.5. We have gone through the manuscript and made sure it is clear which dataset involves LL or FL companions, see line 281, 282, 299, 301, 305, 308, 314-15, 317)

How was the expression of genes such as GaHD1 in the RNA-seq analysis

The RNAseq data for GaGIR1, GaMYB25-like and GaHD-1 are provided in the manuscript (Fig 5 C and 5D, and Fig 3C) as all transcriptional changes are derived from the one RNAseq experiment. We have changed the manuscript to make it clear that this data is derived from the RNAseq dataset. See lines 179-180, 238-239.

Are the genes differentially regulated based on the RNA-seq data are validated using quantitative PCR?

The differentially expressed genes found by RNAseq data have not been validated using QPCR. Since the updated genome sequence of G. arboreum that we used for our analysis is excellent, our RNA sequence coverage high, our analysis methodology stringent and we have 3 biological replicates for each comparison, we feel that this data is superior to QPCR.

The conclusion that GaHD-1 is a candidate gene responsible for the Ga-lintless locus is not validated by functional studies, and needs to be tonned-down.  For example, Line 178-9. Fig. 3C. shows a clear decrease in relative expression, but it is not completely suppressed in A2-106. This difference may not be translated into expression at protein level. 

The reviewer is correct in that the GaHD-1 gene has not been validated functionally in G. arboreum (although it has been validated in G. hirsutum and G. barbadense). So we have toned down our descriptions to indicate that it is a likely candidate gene only. The nature of the mutation found, results in a splicing change (confirmed by cloning and sequencing) resulting in 247 amino acid deletion of the HD-1 protein, which is very likely to render the protein non-functional, but it unlikely to abolish transcription from this gene, so the lower but not abolished transcription observed is consistent with this being the best candidate gene for the lintless phenotype. See lines 182, 378-379.

Specific genes/gene families identified in the RNA-seq are not discussed

As mentioned above we believe the best balance is to investigate and detail biological processes that were altered rather than focusing on specific genes. With the exception being the genes that were potentially candidate genes for the traits we were investigating, eg., GaGIR1, GaMYB25-like and GaHD-1 (that are all mentioned and discussed, and the expression data provided). We provide the reader with all the differentially expressed genes and their gene descriptions in the supplementary tables as well as provide a link to the raw RNA Seq data to allow others to reanalyse the data, therefore readers interested in specific genes can find specific genes and gene families.

Reviewer 2 Report

The article is full of experimental and bioinformatics information of interest primarily to specialists in the field of genetics, genomics, and breeding of cotton. However, it is worthy of publication in a broad-based journal, which is IJMS as an example of a comprehensive study in the field. There are no principal comments on the work. Technical corrections are marked in the attached file.

Author Response

We thank the reviewer for their comments and detailed analysis of the manuscript. We have complied with all the grammatical corrections marked on the manuscript by the reviewer.